# A Review on the Triggers of Pediatric Migraine with the Aim of Improving Headache Education

**DOI:** 10.3390/jcm9113717

**Published:** 2020-11-19

**Authors:** Gaku Yamanaka, Shinichiro Morichi, Shinji Suzuki, Soken Go, Mika Takeshita, Kanako Kanou, Yu Ishida, Shingo Oana, Hisashi Kawashima

**Affiliations:** Department of Pediatrics and Adolescent Medicine, Tokyo Medical University, Tokyo 160-0023, Japan; s.morichi@gmail.com (S.M.); shin.szk@gmail.com (S.S.); soupei59@gmail.com (S.G.); jerryfish_mika@yahoo.co.jp (M.T.); kanako.hayashi.0110@gmail.com (K.K.); ishiyu@tokyo-med.ac.jp (Y.I.); oanas@tokyo-med.ac.jp (S.O.); hisashi@tokyo-med.ac.jp (H.K.)

**Keywords:** headache, migraine, prevention, stress, sleep poverty, alimentation

## Abstract

Although migraines are common in children and adolescents, they have a robustly negative impact on the quality of life of individuals and their families. The current treatment guidelines outline the behavioral and lifestyle interventions to correct common causative factors, such as negative emotional states, lack of exercise and sleep, and obesity; however, the evidence of their effectiveness is insufficient. To create a plan for disseminating optimal pediatric headache education, we reviewed the current evidence for factors correlated with migraine. We assessed three triggers or risk factors for migraines in children and adolescents: stress, sleep poverty, and alimentation (including diet and obesity). While there is a gradual uptick in research supporting the association between migraine, stress, and sleep, the evidence for diet-related migraines is very limited. Unless obvious dietary triggers are defined, clinicians should counsel patients to eat a balanced diet and avoid skipping meals rather than randomly limiting certain foods. We concluded that there is not enough evidence to establish a headache education plan regarding behavioral and lifestyle interventions. Clinicians should advise patients to avoid certain triggers, such as stress and sleep disorders, and make a few conservative dietary changes.

## 1. Introduction

Migraine headaches are common in children, adolescents, and adults worldwide. Its robust negative impact can affect the quality of life of affected individuals in a manner similar to that of childhood cancer, heart disease, and rheumatic disease [1].

The estimated overall mean prevalence of headache is about 50% in the pediatric population according to population-based studies, and the overall mean prevalence of migraine is 9.1% [2,3]. Approximately half of the children with migraine also manifest migraines in adulthood [4,5,6]. Long-term migraine outcomes can be improved in childhood with early diagnosis and multidisciplinary intervention [7,8] comprising both pharmacologic and non-pharmacologic approaches (such as bio-behavioral and biofeedback therapy) [8,9,10]. In adults with migraines, educational initiatives are effective in reducing pain and disability [11,12] and improving the quality of life [13], but apart from this research, there are limited reports of non-pharmacological treatment focused on headache education in adolescents [14,15].

Among these limited reports, a previous retrospective study showed that the participants’ headaches reduced after conservative therapy alone, which consisted of good sleep hygiene, a no-additive diet, and limited sun exposure. The effect was particularly significant in younger children (under 6 years of age) as compared with older children [14]. A cluster-randomized trial involving 1674 adolescent patients with migraine, tension-type headache, or mixed headache assessed preventable risk factors (e.g., physical inactivity, coffee consumption, etc.), stress management, and guided muscle relaxation in the neck and shoulders; and showed a significant reduction in tension-type headaches, but not in migraines [15].

However, there is exceptional evidence in favor of cognitive-behavioral therapy (CBT). According to a recent systematic review, the odds of achieving a 50% or greater reduction in headache activity were 9.11-fold higher after treatment and 9.18-fold higher at the follow-up in patients receiving CBT as compared with those receiving control interventions [16]. Therefore, some researchers opine that CBT should be the first-line treatment for pediatric migraine due to the high evidence level and absence of associated harm [17]. However, CBT is not always practicable because psychological treatment may not be acceptable for some patients. It is costly, time-consuming, and may be effective only in older children who are capable of metacognition [18,19]. CBT may be available in only highly specialized facilities, which makes it difficult to spread the use of CBT widely.

Compared to the increasing volume of research in support of CBT––headache education, a basic and important non-pharmacological form of treatment––may get neglected. Although the current treatment guidelines focus on behavioral and lifestyle interventions to correct the factors commonly associated with migraine (negative emotional states, obesity and high body weight, infrequent exercise, and lack of sleep) [20], the focus on dispensing headache and migraine education is inadequate, making this an area that requires extensive research [21]. Without quality evidence on how proper headache education can improve outcomes, preventing migraines will remain challenging for many children and adolescents [21].

### Study Objective

The goal was to establish standardized headache education that can be implemented in pediatric patients with migraine who are unable to receive CBT or in institutions that are unable to provide CBT. In order to do this, we need to be certain about what is good and what is not good for a child with migraine. Thus, we reviewed the current evidence on factors correlated with migraine and focused on the following factors: stress, sleep poverty, and alimentation (including diet and obesity)—all of which are known to be the most potent risk factors for migraine attacks in children and adolescents. In order to summarize the knowledge required for clinicians and healthcare workers to provide proper headache education, we reviewed the link between lifestyle, behavioral triggers, and migraine.

## 2. Summary of the Literature Search

This narrative literature review is based on studies identified in the PubMed database published until June 2020, with no date limits. Studies were identified using various search terms: “migraine,” “primary headache,” “children, ” “adolescent,” “non-pharmacological treatment,” “education,” “life style,” “management,” “trigger,” “stress,” “sleep,” “alimentation,” “diet,” “food,” “obesity,” “quality of life,” “psychiatric disorders,” and “adverse experiences.” The included studies were observational studies, randomized controlled trials, systematic reviews, meta-analyses, and narrative reviews. Only human studies published in English were considered. Individual case reports were excluded. Articles were also added based on the authors’ knowledge of the area. Generally, articles on children and adolescents younger than 18 years of age were selected, but adult subjects were cited as appropriate when there was little evidence for children and adolescents.

## 3. Prevalence of Trigger Factors in Migraine

Although there are several proposed triggers for migraine attacks, recent literature on the childhood triggers of migraine is limited, as shown in Table 1 [22,23,24]. A retrospective clinical study assessed the prevalence of possible migraine trigger factors in 102 children and adolescents and found that the most frequently reported trigger factor was stress (75.5% of patients), followed by lack of sleep (69.6%), warm climate (68.6%), and video games (64.7%) [23]. The study also showed that the mean number of migraine triggers reported per subject was 7, and the mean time elapsed between exposure to a trigger factor and onset of the attack was 0–3 hours in 86% of patients. Later, to confirm these results, the same group conducted a prospective clinic-based study over a 3 month period on 101 pediatric patients; wherein each patient reported at least 1 trigger (range: 1-14; median: 3) with a total number of 532 attacks. The study demonstrated that lack of sleep (51.4%), stress (44.6%), warm climate (41.9%), noise (32.4%), and excitation (29.7%) emerged as triggers and that the period between trigger exposure and attack onset was 0–3 hours in 67.6% of patients [24]. Even if the number of triggers was lesser, the same results that implicated the four most-frequent triggers (lack of sleep, stress, hot weather, and noise) were observed in both studies [23,24]. Another questionnaire survey described the following causative factors for migraine attacks: bad sleep (32.9%), emotional distress (25.7%), intense noise or light (7.1%), and weather conditions (12.9%) [22].

### 3.1. Trigger factors

#### 3.1.1. Stress

The most problematic and common trigger for pediatric migraines is stress [23,24]. Several reports have suggested that stressful home [25,26] and school [27,28,29] environments aid migraine attacks. The brain structure becomes disrupted by stress due to multifactorial and epigenetic processes that alter gene expression [30,31]. If registered biologically during the early developmental stages, stress can lead to physiological changes that may increase a child’s susceptibility to other stress-related behavioral disorders, such as depression and anxiety, later in life [31,32].

#### 3.1.2. Psychiatric Disorders

A cross-sectional national Canadian survey of 61,375 participants aged 12-19 years detected a consistent association between migraine and mood/anxiety disorders [33]. A similar survey demonstrated that the only risk factor significantly associated with a headache-related disability was depression, which was also positively correlated with stress, sleep disturbance, and anxiety [34]. In another longitudinal study, 12.5% of children with migraine were categorized as depressed as compared with 5.9% of children without headache, although there was no significant relationship between depressive symptoms and migraine [35].

Migraine and psychiatric conditions, such as mood disorders and anxiety, might be comorbid [36,37,38,39]. However, behavioral comorbidities have not been observed in the majority of pediatric migraine patients. Currently available screening tools make it difficult to distinguish between the characteristics of migraine and those of psychiatric illness [36,37,40,41]. A recent longitudinal cohort study on 2313 respondents who were followed prospectively from 0–1 year of age (baseline) until 14–15 years detected depression- and anxiety-mediated relationships between family dysfunction, punitive parenting, and parental depressive symptomatology only in patients with migraine. The presence of these family-level stressors early in life was associated with a higher likelihood of adolescent migraine and increased symptoms of depression and anxiety in late childhood [25]. Therefore, if psychiatric comorbidities are detected in a patient, they should be addressed in a timely fashion and managed appropriately to prevent the development of migraines in the future.

#### 3.1.3. Adverse Childhood Experiences

Adverse childhood experiences, such as child abuse, are strongly associated with recurrent migraines in adulthood [42,43,44,45,46,47,48,49,50]. A survey of 10,358 men and 12,638 women revealed that a greater number of childhood adversities was associated with increasing odds of migraine occurrence [48]. A recent systematic review reported that a high-functioning family is a protective factor; it is associated with better management of chronic migraine pain in children [50]. Undoubtedly, creating optimal and healthy environments at home and school can bolster an ongoing course of migraine treatment.

### 3.2. Sleep and Migraine

Literature on the relationship between primary headaches and sleep disturbances is growing steadily, as is evidenced by several recent questionnaires that assess sleep difficulties [51,52,53,54,55,56,57,58,59]. Migraine and sleep disturbance are certainly linked, and a bidirectional relationship between them has been described in childhood [52,59,60]. In other words, migraine might be a result of underlying sleep disorders or it might trigger further sleep disturbances.

#### 3.2.1. High Prevalence of Various Sleep Disorders

A high range of sleep disturbances has been reported in children with migraine, as shown in Table 2 [52,56,61]. A questionnaire study consisting of 118 children aged 2–12 years demonstrated that insufficient total sleep (42%), bruxism (29%), co-sleeping with parents (25%), and snoring (23%) were indicative of migraine onset [52]. Another questionnaire study involving 69 adolescents aged 13–17 years with headache (90% migraine; 10% tension-type headache) indicated that insufficient total sleep (65.7%), daytime sleepiness (23.3%), difficulty in falling asleep (40.6%), and night walking (38%) were major migraine indicators [56]. According to both studies, sleep deprivation was the most common cause of migraines in children and adolescents [52,56]. The risk factors that were most sensitive to impending sleep disorders were daytime sleepiness and habitual snoring [57]. In an objective evaluation with polysomnography, migraine patients aged 5-14 years tended to snore (66%), had problems in sleep onset and maintenance (25%), and experienced excessive daytime sleepiness (20%). Moreover, 40% of patients had obstructive sleep apnea, 27% had insomnia, 15% had periodic limb movement disorder, and 6% had a central disorder of hypersomnolence [61]. Although a few other reports have suggested that there is no remarkable difference in the type of sleep disorder and migraine onset [54,55], there is no doubt that overall, children with migraines are more likely to develop sleep disorders.

#### 3.2.2. Poor Sleep Quality in Children with Migraine

Children with migraine have poorer sleep quality; this has been established in questionnaire studies [22] and under objective monitoring [61,62,63]. Polysomnographic studies have shown that sleep disorders occur in patients with migraine [61,62] and that migraines may reduce the quality of sleep. An actigraphy analysis of 18 migraine patients aged 8–12 years old found only a slight prolongation of sleep onset latency in the participants [62]. However, a polysomnographic study showed that 53 of 90 headache patients aged 5–19 years with severe and chronic migraines exhibited disrupted sleep architecture, reduced rapid eye movement, and slow-wave sleep [63]. Another similar study demonstrated changes in sleep architecture (increased non-rapid eye movement and decreased slow-wave sleep) [61].

#### 3.2.3. Features of Headache and Sleep Disorders

Several questionnaire studies have suggested that the frequency and intensity of migraine headaches can predict the occurrence of sleep disorders, such as anxiety, parasomnia, and bedtime resistance, in children and adolescents [56,64]. A recent prospective cohort study comprising patients aged 10-18 years old with migraine or probable migraine and without daily sleep complaints demonstrated that headache intensity and timing of headache onset were predictive of sleep disturbances. Another study detected a significantly positive correlation between sleep disturbances and daily headache disability scores [59]. Additionally, in both a questionnaire-based survey [55] and a polysomnographic study [63], more severe migraines indicated severe sleep disturbances. These results suggest that migraine headaches correlate with the severity of sleep disorders to a substantial degree.

#### 3.2.4. Improving Quality of Sleep Can Improve Migraines

Although it is not established whether migraines reduce the quality of sleep or if they are a result of poor sleep quality, improving the quality of sleep can decrease the occurrence of migraines [57,65]. In a previous clinical trial, the parents of migraine patients were requested to enforce the following guidelines once a week: bedtime later than 11 p.m.; wake-up time later than 8 a.m.; a nap during daytime; irregular schedule (bedtime and wake-up time to vary by more than 1 hour on school days); regular intake of cola, tea, coffee, chocolate, or similar substances late in the afternoon or evening; and facilitating the need to drink fluids or take drugs to fall sleep. After the study period, the poor-sleep-hygiene group was compared with a group that was educated on healthy sleep habits; wherein the researchers observed that the mean duration of a migraine attack at the 3- and 6-month follow-up appointments were significantly reduced in the educated group, while the poor-sleep group showed an insignificant initial reduction in the attack duration. They found a significant reduction in headache frequency only in the educated group [65].

Daytime sleepiness and habitual snoring are common in pediatric patients with migraine, and improving snoring can reduce daytime sleepiness [57]. Along with the above-mentioned study, this result suggests that improving the quality of sleep through healthier habits (without relying on prophylactic medication) can reduce the frequency of migraine headaches. Notably, sleeping itself is effective in relieving headaches or terminating migraines [14,66,67]. Sleep disruption represents a potentially modifiable vulnerability to migraines and vice versa. Regardless of whether it is the trigger or effect of migraine, the two are closely related and could be thought of as an expression of a common pathogenic process [68].

Conventional clinical wisdom, sleep assessment, and adjustment in sleep habits will always be recommended for children suffering from migraines, in addition to which structured sleep hygiene rules can improve both headaches and sleep in the long run.

### 3.3. Alimentation

Certain foods, such as chocolate, caffeine, milk, and cheese, are common triggers of migraine attacks [69,70]. Dietary habits play an important role in precipitating headaches in children and adolescents with migraine. These triggers might affect different stages of the migraine process by influencing the release of serotonin and norepinephrine, causing vasoconstriction or vasodilatation, or stimulating the trigeminal ganglia, brainstem, and cortical neuronal pathways [71].

#### 3.3.1. Prevalence of Dietary Triggers in Migraine

Previous literature reviews have revealed that 12–60% of subjects in different studies reported specific foods as a trigger for adult migraines, with many subjects reporting more than one dietary trigger [72]. In a prospective study on potential triggers in individuals with pediatric migraine, 38.1% were diet-related (caffeine: 7.7%, nuts: 7.4%, citrus fruits: 6.1%, monosodium glutamate: 6.1%, artificial sweetener: 4.0%, nitrates: 3.4%, chocolate: 2.5%, cheese: 0.9%) [73]. In 1979, a questionnaire survey of 120 children with migraine demonstrated that diet-related causative factors were present in 70% of children, with specific foods comprising 38% of the assessed factors (chocolate 17%, cheese 16%, citrus fruits 5%,) and fasting accounting for 41% [74]. In a retrospective clinical study comprising 102 children and adolescents with migraine, the percentage of migraine patients reporting a particular food or drink that triggered their attack was 32.3% (chocolate: 11.8%, colas: 8.8%, soft drinks: 3.9%, citrus fruits: 3.9%, cheese: 3.9%) and 30.4% of the individuals were fasting [23].

Evaluating the role of diet in migraine is complex because multiple triggers and variables can modify the pain threshold in an individual, and this factor is frequently neglected in favor of preventive drug therapy for pediatric patients [75]. Although chocolate is the most frequent dietary trigger for migraines in children [23,74], a recent review article found insufficient evidence of chocolate being described as a definitive migraine trigger [76].

#### 3.3.2. Randomized Control Trial of Dietary Therapy

There are extremely limited randomized control trials investigating the role of diet in children with migraines [77,78]. A prospective study on 88 children with migraines assessed the effect of an oligoantigenic diet—an elimination food plan designed to identify and eliminate a large range of potentially triggering foods—on migraines. The causative foods were identified by sequential reintroduction and the migraine-provoking food items were evaluated by a double-blind controlled trial in 40 children. Most children (93%) showed complete or significant improvement while on the diet; however, 8 patients relapsed when one or more foods were reintroduced [77]. A small randomized trial consisting of 39 participants with pediatric migraine investigated the efficacy of a high-fiber diet alone versus a high-fiber diet coupled with the elimination of foods high in vasoactive amines; there was a significant decrease in the number of headaches in both groups, but there was no significant between-group difference [78].

Current research on adults shows that high folate diets, low-fat diets, low-sodium diets, foods high in high omega-3 fatty acids, and ketogenic diets are effective in reducing migraines [79,80]. However, there is a lack of scientific evidence to support these diets as a therapeutic recommendation [75]. Even chocolate lacks sufficient evidence of being labeled a proper trigger, as per a recent systematic review [76]. Certain dietary interventions may improve clinical outcomes in some individuals with migraines; these associations need to be confirmed by high-quality longitudinal studies [75].

We consider that it is imperative to counsel patients and their parents/caretakers to avoid fasting or skipping meals and encourage a well-balanced diet, along with educating them on good practices to avoid obvious triggers.

#### 3.3.3. Obesity

Obesity has a proven association with the pathogenesis of migraine [81]. Current research suggests that migraine and obesity may be linked through inflammatory mediators released by adipose tissue [81,82,83].

A high prevalence of obesity (17.7–56%) in individuals with migraine has been found in several previous studies [2,84,85,86]. As suggested by a large-scale retrospective study on 181 children and adolescents, a diagnosis of migraine (and not a tension-type headache) was significantly associated with individuals who were at risk of being overweight, particularly women [86]. Another large study involving 124 pediatric migraine patients showed that obese patients had more frequent migraine attacks than overweight and normal-weight patients, and described a significant positive correlation between the relative body mass index and number of attacks [85]. Losing weight might contribute to the reduction of migraine frequency, intensity, and disability in obese children [8,87]. A controlled clinical trial involving 40 adult migraineurs demonstrated that consistent and regular exercise led to reduced pain severity, frequency, and duration, possibly due to increased production of nitric oxide [88].

Although the current evidence on the efficacy of weight-loss interventions for migraine is limited, maintaining a healthy weight may be more effective in treating migraines than trying to eliminate uncertain migraine triggers.

## 4. Discussion

To summarize, the knowledge required by clinicians to dispense proper headache education and counsel their patients on behavioral and lifestyle interventions, we reviewed the link between migraine and its common triggers. We assessed the following factors: stress, sleep poverty, and alimentation. To date, there is no concrete evidence that strongly indicates the effectiveness of headache education.

Stress within the family and in school environments is linked to migraines in children and adolescents. These stresses can exacerbate a migraine, which in turn reduces further functioning [29,34], impairs the health-related quality of life [28,89], and affects relationships with colleagues [90]. Ensuring healthy family dynamics as part of routine consultations may reduce unnecessary stressful burdens on children and adolescents and improve their physical complaints [25]. Understanding that migraines are associated with family [91] and building better relationships to form a solid support system will reduce migraines in affected children. Proper and timely assessment of psychiatric comorbidities may alleviate the patient’s added stress due to the connection between migraines and psychiatric comorbidities, such as depression and anxiety. Psychiatric comorbidities can deteriorate a home environment, and a worsening home situation increases the likelihood of future migraine development.

The association between sleep and migraine is supported by evidence, barring the conundrum of whether sleep is a trigger or an effect of migraine. The assessment and adjustment of sleep habits are always recommended for children suffering from migraines. According to the National Sleep Foundation, the recommended sleep times are 9–12 hours for children (6–12 years) and 8–10 hours for teenagers (13–18 years) [92]. After adjusting the time of sleep and maintaining good sleep hygiene, children with persistent migraines should be screened by polysomnography to identify possible sleep disorders. We conjecture that sleep education plays a significant role in the success of migraine treatment.

The current evidence on dietary triggers of migraine is limited, but several large population-based studies have exhibited a negative view of the efficacy of restrictive diets or weight-loss interventions [93,94,95]. Rather than avoiding diet-related triggers, it is more important to avoid skipping meals and instead eat a well-balanced diet. The Headache Diaries prospectively gathers information on potential triggers, including behavioral, dietary, and environmental factors, to identify individual triggers [73]. However, completely avoiding all potential headache triggers is unlikely because of their diversity, and attempting to do so could result in a very restricted lifestyle [96]. Limiting all triggers can be stressful and harmful, far outweighing any potential benefits and lowering the threshold for migraine development [17]. Present trends in migraine management are thought to be more important in managing triggers than avoiding them [97,98]. We recommend that physicians should not solely advise children and their caregivers to ‘avoid stress’, but also counsel them to respond to incentives through instruction and ‘coping mechanisms’, similar to those for adult migraine patients [99,100]. A recent observational study showed that pediatric migraine patients practiced fewer pain-coping strategies [33]; we advocate that children with migraines need to be taught child-appropriate coping methods. The role of the parents/caretakers around the child with a migraine is also important. By creating optimal and healthy environments, they need to explore what kind of lifestyle is appropriate for the affected child. The child’s lifestyle should be tailored to his/her individual needs, including getting regular exercise, eating a proper diet, and sleeping on a regular schedule (Figure 1). And they need to learn the mechanisms of migraines, the fact that a migraine is a disorder of the brain that involves altered sensory processing [101], and that lifestyle habits can decrease the risk of developing a headache attack.

While it is critical to encourage good sleep hygiene and ensure healthy family relationships, a physician should also emphasize the importance of a balanced diet and should not advise more food restrictions than necessary. To prevent migraines, affected children and their parents/caretakers must manage stress, sleep, and diet—overall following a balanced lifestyle; this is essential for the health of the child and the family.

Future larger, prospective studies are needed to design a standardized educational and cognitive treatment manual that will contribute to the development of pediatric migraine treatment.

## Figures and Tables

**Figure 1 jcm-09-03717-f001:**
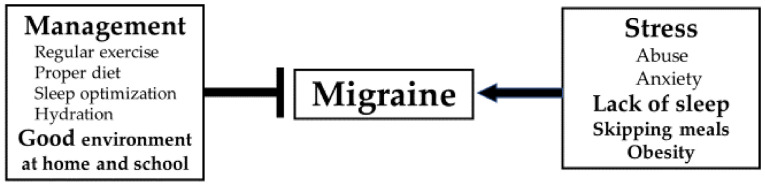
Lifestyle management for migraines.

**Table 1 jcm-09-03717-t001:** The prevalence of trigger factors in pediatric patients with migraine.

References	Study Design	Number of Patients with Migraine	Age (Range) (Years)	Migraine Trigger Factors (%)					
				Stress	Lack of Sleep	Weather Condition	Video Games	Intense Noise or Light	Excitation
Neut D. et al., 2012 [23]	Retrospective Clinical Study	102	7–16	75.5	69.6	68.6	64.7	ND	ND
Solotareff L. et al., 2017 [24]	Prospective Clinic-Based Study	101	5–17	44.6	51.4	41.9	ND	32.4	29.7
Bruni O. et al., 2008 [22]	Retrospective Clinical Study	70	8–15	25.7	32.9	12.9	ND	7.1	ND

ND: not described.

**Table 2 jcm-09-03717-t002:** The prevalence of various sleep disorders in pediatric patients with migraine.

References	Study Design	Number of Patients with Migraine	Age (Range) (Years)	Sleep Disorders (%)							
				Insufficient Total Sleep	Bruxism	Co-Sleeping with Parents	Snoring	Daytime Sleepiness	Difficulty with Falling Asleep	Night Walking	Having Onset and Maintenance
Miller V.A. et al., 2003 [52]	Retrospective Clinical Study	118	2–12	42	29	25	23	ND	ND	ND	ND
Gilman D.K. et al., 2007 [56]	Retrospective Clinical Study	69 ^§^	13–17	65.7	ND	ND	ND	23.3	40.6	38	ND
Armoni Domany K. et al., 2019 [62]	Retrospective Review	185	5–14	ND	ND	ND	66	20	ND	ND	25

§: It includes tension-type headache. ND: not described.

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
