# Peer review of "A Review on the Triggers of Pediatric Migraine with the Aim of Improving Headache Education"

_jcm, 2020, doi:10.3390/jcm9113717_

Round 1
Reviewer 1 Report
The authors review available evidence on migraine triggers in the pediatric population and discuss consequences for counselling of the pediatric migraine population. The main triggers are food and various sleep disturbances. However, little is known about the effect on migraine frequencies itself if those triggers were avoided.
However, there are some points to discuss:
Current trend in migraine therapy is more focusing on trigger management than on avoiding triggers (for example: Klan et al. Headache. 2019 May;59(5):741-755; Martin PR. Headache 2020 Jul 10. doi: 10.1111/head.13901. Online ahead of print.). A simple “do not do and do not eat-list” might be not very successful in the pediatric population.
Little is mentioned about headache education in general. The role of the parents needs to be considered.
The authors mentioned that CBT might be not acceptable for some patients, that’s true but rather its not available outside highly specialized care.
A requirement before providing life style recommendation on the pediatric migraine population (and the parents) is proven efficacy of this recommendation. A standardized education and cognitive treatment manual which is evaluated in a prospective study is needed.
Author Response
Responses to the Reviewers’ Comments
Reviewer: 1
We appreciate the time and effort you have dedicated to providing this insightful feedback. We have revised our manuscript in accordance with your comments. All suggested revisions, as well as additional ones to improve the language of the manuscript, are indicated in underline text. We hope that, with these revisions, our manuscript will be suitable for publication.
Comment 1: Current trend in migraine therapy is more focusing on trigger management than on avoiding triggers (for example: Klan et al. Headache. 2019 May;59(5):741-755; Martin PR. Headache 2020 Jul 10. doi: 10.1111/head.13901. Online ahead of print.). A simple “do not do and do not eat-list” might be not very successful in the pediatric population.
Response: We agree with you. We have cited the two papers you referred to and made the following changes (page 10, lines 143).
“Present trends in migraine management are thought to be more important in managing triggers than avoiding them [97,98].”
Comment 2: Little is mentioned about headache education in general. The role of the parents needs to be considered.
Response: Thank you for your valuable input. We have added the following text in the Discussion (page 11, lines 150). In addition, we have added a Figure 1 of lifestyle management for migraines (page 11, lines 164). If the referee deems it unnecessary, we will of course leave it out.
“The role of the parents/caretakers around the child with a migraine is also important. By creating optimal and healthy environments, they need to explore what kind of lifestyle is appropriate for the affected child. The child's lifestyle should be tailored to his or her individual needs, including getting regular exercise, eating a proper diet, and sleeping on a regular schedule. And they need to learn the mechanisms of migraines, the fact that migraine is a disorder of the brain that involves altered sensory processing [101], and that lifestyle habits can decrease the risk of developing a headache attack.
Figure 1. Lifestyle management for Migraine
Comment 3: The authors mentioned that CBT might be not acceptable for some patients, that’s true but rather its not available outside highly specialized care.
Response: We agree with you and have revised the text accordingly (page 2, lines 54).
“CBT may be available in only highly specialized facilities, which makes it difficult to spread the use of CBT widely.”
Comment 4: A requirement before providing lifestyle recommendation on the pediatric migraine population (and the parents) is proven efficacy of this recommendation. A standardized education and cognitive treatment manual which is evaluated in a prospective study is needed.
Response: We agree with you. Reports on prospective headache education are scarce, and the level of evidence is currently lacking. The following was noted at the end of the Discussion (page 11, lines 161).
“Future larger, prospective studies are needed to design a standardized educational and cognitive treatment manual that will contribute to the development of pediatric migraine treatment.”

Reviewer 2 Report
This is a useful narrative scoping review to formulate a headache behavioral education for pediatric migraine sufferers. The review needs better structuring, currently, it looks like a chapter out of a textbook. There are several guidelines on how to write a narrative or scoping review. E.g. https://training.cochrane.org/resource/scoping-reviews-what-they-are-and-how-you-can-do-them
The review needs a better reformatting to indicate that the focus is on behavioral and lifestyle approaches. Please clarify on the study objective and question the review tries to address. It will help to describe how the scoping review was made, what literature sources and search methods were utilized? Try to include familial/care givers education especially in the case of younger pediatric age group. Include a paragraph for exercise approaches.
Author Response
Referee 2
We would like to thank you for carefully reviewing our manuscript and for providing valuable comments. We have revised our manuscript in accordance with your comments. All suggested revisions, as well as additional ones to improve the language of the manuscript, are indicated in underline text. We hope that, with these revisions, our manuscript will be suitable for publication.
Comment 1: The review needs a better reformatting to indicate that the focus is on behavioral and lifestyle approaches. Please clarify on the study objective and question the review tries to address. It will help to describe how the scoping review was made, what literature sources and search methods were utilized?
Response: As per your suggestion, to clarify the goals, we have changed the formas by including the following in the abstract and text.
Abstract (page 1, lines 16).
“To create a plan for disseminating optimal pediatric headache education, we reviewed the current evidence for factors correlated with migraine.”
“1.1. Study objective (page 2, lines 69).
The goal was to establish standardized headache education that can be implemented in pediatric patients with migraine who are unable to receive CBT or in institutions that are unable to provide CBT. In order to do this, we need to be certain about what is good and what is not good for a child with migraine. Thus, we reviewed the current evidence on factors correlated with migraine and focused on the following factors: stress, sleep poverty, and alimentation (including diet and obesity)––all of which are known to be the most potent risk factors for migraine attacks in children and adolescents. In order to summarize the knowledge required for clinicians and healthcare workers to provide proper headache education, we reviewed the link between lifestyle, behavioral triggers, and migraine.”
- Summary of the literature search (page 2, lines 79).
“This narrative literature review is based on studies identified in the PubMed database published until June 2020, with no date limits. Studies were identified using various search terms: “migraine,” “primary headache,” “children,” “adolescent,” “non-pharmacological treatment,” “education,” “life style,” “management,” “trigger,” “stress,” “sleep,” “alimentation,” “diet,” “food,” “obesity,” “quality of life,” “psychiatric disorders,” and “adverse experiences.” The included studies were observational studies, randomized controlled trials, systematic reviews, meta-analyses, and narrative reviews. Only human studies published in English were considered. Individual case reports were excluded. Articles were also added based on the authors’ knowledge of the area. Generally, articles on children and adolescents younger than 18 years of age were selected, but adult subjects were cited as appropriate when there was little evidence for children and adolescents.”
Comment 2: Try to include familial/care givers education especially in the case of younger pediatric age group. Include a paragraph for exercise approaches.
Response: Response: Thank you for your valuable input. We have added the following text in the Discussion (page 11, lines 150). In addition, we have added a Figure 1 of lifestyle management for migraines (page 11, lines 164). If the referee deems it unnecessary, we will of course leave it out.
“The role of the parents/caretakers around the child with a migraine is also important. By creating optimal and healthy environments, they need to explore what kind of lifestyle is appropriate for the affected child. The child's lifestyle should be tailored to his or her individual needs, including getting regular exercise, eating a proper diet, and sleeping on a regular schedule. And they need to learn the mechanisms of migraines, the fact that migraine is a disorder of the brain that involves altered sensory processing [101], and that lifestyle habits can decrease the risk of developing a headache attack.
Figure 1. Lifestyle management for Migraine
